# Hypergravity Increases Blood–Brain Barrier Permeability to Fluorescent Dextran and Antisense Oligonucleotide in Mice

**DOI:** 10.3390/cells12050734

**Published:** 2023-02-24

**Authors:** David Dubayle, Arnaud Vanden-Bossche, Tom Peixoto, Jean-Luc Morel

**Affiliations:** 1CNRS, INCC, UMR 8002, Université Paris Cité, F-75006 Paris, France; 2INSERM, SAINBIOSE U1059, Mines Saint-Etienne, Université Jean Monnet Saint-Étienne, F-42023 Saint-Étienne, France; 3University Bordeaux, CNRS, INCIA, UMR 5287, F-33000 Bordeaux, France; 4University Bordeaux, CNRS, IMN, UMR 5293, F-33000 Bordeaux, France

**Keywords:** blood–brain barrier, permeability, centrifugation, hypergravity, dextran, antisense oligonucleotide

## Abstract

The earliest effect of spaceflight is an alteration in vestibular function due to microgravity. Hypergravity exposure induced by centrifugation is also able to provoke motion sickness. The blood–brain barrier (BBB) is the crucial interface between the vascular system and the brain to ensure efficient neuronal activity. We developed experimental protocols of hypergravity on C57Bl/6JRJ mice to induce motion sickness and reveal its effects on the BBB. Mice were centrifuged at 2× *g* for 24 h. Fluorescent dextrans with different sizes (40, 70 and 150 kDa) and fluorescent antisense oligonucleotides (AS) were injected into mice retro-orbitally. The presence of fluorescent molecules was revealed by epifluorescence and confocal microscopies in brain slices. Gene expression was evaluated by RT-qPCR from brain extracts. Only the 70 kDa dextran and AS were detected in the parenchyma of several brain regions, suggesting an alteration in the BBB. Moreover, Ctnnd1, Gja4 and Actn1 were upregulated, whereas Jup, Tjp2, Gja1, Actn2, Actn4, Cdh2 and Ocln genes were downregulated, specifically suggesting a dysregulation in the tight junctions of endothelial cells forming the BBB. Our results confirm the alteration in the BBB after a short period of hypergravity exposure.

## 1. Introduction

Astronauts are exposed to successive phases of hypergravity phases during the takeoff and landing of spaceflights and due to microgravity in space. The most important and earliest reported symptom, related to days 1–3 of the spaceflight, is space motion sickness due to vestibular dysfunction [1,2,3,4,5,6,7]. The fluid shift is the shift in the distribution of human body fluids due to microgravity exposure. It was proposed to be responsible for space motion sickness. Several ground devices and protocols were rapidly developed to reproduce this phenomenon, such as centrifugation and parabolic flights [8,9]. Moreover, the decreases in plasma volume and cardiac performance, and the increase in intracranial blood pressure participate in vascular deterioration, as recently reviewed [10,11,12]. Furthermore, the alterations in gravity induce cardiovascular adaptations via the modifications in endothelial and smooth muscle vascular cell functions [13,14,15,16,17]. It is noticeable that the effects of centrifugation are only partially described in humans [18,19,20,21,22]. Like during microgravity exposure, hypergravity exposure, from 1.5 to 5 g, affects the vestibular functions [23,24,25,26] and modifies gene expression in the brain [27,28,29] and cognitive performances [30,31,32]. The use of hypergravity by centrifugation is required to qualify the biological effects of space motion sickness.

Likewise, centrifugation, close to 2× *g*, is also proposed as a countermeasure against the deleterious effects of microgravity seen in humans [33,34]. Therefore, before the exposure of humans to centrifugation, it is important to study its biological impacts.

The cerebral blood vessels are crucial in brain functions regarding oxygen supply and exchanges of nutrients and wastes. The endothelial cells of brain capillaries are organized to form the blood–brain barrier (BBB), assuming the fine-tuning of these exchanges to maintain brain homeostasis [35,36]. The efficacy of the BBB is regulated by the nychthemeral rhythms [37,38,39,40]. BBB alterations are clearly implicated in stroke and neurodegenerative disorders [41,42,43,44]. 

Gravity changes are able to modify endothelial cell functions [45]. Many in vitro models have been developed to reproduce the BBB [46], and experiments that exposed endothelial cells to gravity modifications revealed miscellaneous results, as reviewed [47]. Depending on hypergravity levels from 3 g to 20 g, endothelial cells modify their gene expression, angiogenesis, cytoskeleton architecture and tube formation [48,49,50,51]. Moreover, in devices that reproduce the barrier function, the effects of short-term exposure to hypergravity remain unclear. In fact, exposure (2 g and 4 g) increases the barrier efficacy, shown by resistance measurements of the endothelial cell culture [52], whereas a higher level (10 g) decreases it, as shown by the increase in fluorescent molecules passing through the culture monolayer [53].

The effects of hypergravity on the capacity of endothelial cells to form a barrier in vitro are insufficient to interpret the modifications in the BBB observed in vivo. More information should be collected in vivo. 

In mice exposed to hypergravity at 2 g for 24 h, we measured the transit through the BBB of different fluorescent molecules with different sizes, such as dextrans and antisense oligonucleotides (AS). We also investigated the regulation of expression of genes involved in junctions between endothelial cells.

## 2. Materials and Methods

### 2.1. Animals and Centrifugation

In accordance with the principles of the European community, the experimental protocols were validated by the local ethics committee (CEEA-Loire, APAFIS #38819), the animal welfare committee of PLEXAN (PLateforme d’EXpérimentations et d’ANalyses, Faculty of Medicine, Université Jean Monnet, Saint Etienne, France, agreement n°42-18-0801) and the French Ministry of Research. In this study, 86 male C57BL/6JRJ mice (8 weeks old, 22.5 ± 0.1 g, Janvier Labs, France) were used. The animals were housed (3 mice per cage) in standard conditions (22 °C, humidity 55%; 12 h/12 h day/night cycle; unlimited access to food and water). They were familiarized with the centrifugation room the week before the experiments and monitored by video in the centrifuge. In order to expose all the animals to the same environmental conditions, the mice were centrifuged at 2× *g* for 24 h, and the control mice in normogravity at 1 g for 24 h were placed simultaneously in the experimental room. The centrifugation protocol was detailed in our previous publication. 

### 2.2. In Vivo Injection of Antisense Oligonucleotide and Dextrans

All the fluorescent molecules were diluted in saline solution (sodium chloride 9 g/L) and retro-orbitally injected in the blood, under isoflurane anesthesia (5%). In our hands, this route of administration is safer (more rapid, efficient and reproducible) than other routes of i.v. administration. Sham mice were injected with vehicle solution. Mice received only one injection with one fluorescent tracer. Phosphorothioate antisense oligonucleotide directed against angiopoietin-2 (Angpt2, named AS, GCG-TTA-GAC-ATG-TAG-GG, 6084.9 g/mol, Eurogentec) was coupled to 5-carboxyfluorescein (excitation: 492 nm, blue light; emission: 518 nm, green light) and injected (18 mg/Kg). Fluorescein isothiocyanate-dextrans D40, D70 and D150 (FD40-100MG, FD70S-100MG, FD150S-1G, respectively, Sigma-Aldrich, St. Louis, MI, USA) were solubilized in vehicle (2× *g*/100 mL to be injected retro-orbitally at 150 mg/Kg, near 200 µL/mouse). Fluorescein isothiocyanate-dextrans were maximally excited at 490 nm (blue) and maximally emitted at 525 nm (green).

### 2.3. Collection of Biological Samples 

Mice were randomly killed by lethal intraperitoneal injection of sodium pentobarbital (Euthasol, 175 mg/Kg, i.p.), within 2 h after stopping the centrifuge. Before intracardiac perfusion, a catheter was introduced in the right atrium, and blood samples were collected and placed in microtubes. Finally, mice were perfused intracardiacally (5 mL/min) with 30 mL of phosphate-buffered saline (0.01 M PBS, pH: 7.4) to discard blood cells and residual fluorescence of the injected tracers into vessel lumen. This step was followed by 30 mL formalin solution (10%, Merck, HT501128) to fix the tissues. Brains and the left lobe of livers were dissected and post-fixed for 24 h in a formalin solution at room temperature, placed for 48 h in a 30% sucrose–PBS solution at 4 °C and cryopreserved before being sliced.

### 2.4. Corticosterone Assay

The microtubes containing blood samples were centrifuged (10 min at 2000× *g*) and 20 µL of serum was collected. Some serum samples were excluded due to hemolysis. The others (n = 60) were used for corticosterone assay (ELISA kit, K014, Arbor Assays, Ann Arbor, MI, USA), following the protocol of the supplier. 

### 2.5. Histology

Using a freezing microtome (frigomobil, Reichert-Jung), coronal sections of the brain (40 μm thick) were made. Olfactory bulbs were removed, and 3.2 mm after beginning the rostro-caudal slicing, the new slices were collected and placed individually in 48-well plates. To ensure reproducibility, we anatomically selected three similar brain slices for each mouse. Using a binocular device, the slices corresponding to interaural 1.98 mm; Begma −1.82 mm of the Atlas of the mouse brain in stereotaxic coordinates [54] were retained. Indeed, the medial habenular nuclei and mammillothalamic tract were anatomical landmarks, as well as the form and volume of the hippocampus. In the same manner, the left lobes of the liver were sliced (40 µm), and three slices per mouse were mounted. All the floating sections were incubated for 10 min in DAPI (4′,6-diamidino-2-phenylindole, 1:250,000, Interchim, Mannheim, Germany) and rinsed twice in PBS (10 and 20 min, respectively). Finally, they were mounted on glass slides (Superfrost) with a handmade medium based on Mowiol. All slices were DAPI-labeled and mounted on the same day. Slices presenting red blood cells in capillaries in ROI were excluded to reduce experimental bias [55].

### 2.6. Image Acquisition 

The fluorescence of labeled brain slices was observed by confocal microscopy (SP5, Leica Microsystems, Wetzlar, Germany) and the slide scanner Nanozoomer (2.0 HT, Hamamatsu Photonics, Shizuoka Prefecture, Japan). The Nanozoomer contains a fluorescence imaging module using objective UPS APO 20X NA 0.75 combined with an additional lens 1.75X. Virtual slides were acquired with a TDI-3 CCD camera. The fluorescent acquisitions were conducted with a mercury lamp (LX2000 200W—Hamamatsu Photonics, Massy, France), and the set of filters adapted for DAPI and FITC/FAM fluorescence were usable for both fluorescein isothiocyanate-dextrans and 5-carboxyfluorescein antisense oligonucleotide. The DAPI labeling, revealing the double strain of DNA in the cell nuclei, was used for the automated focus required for Nanozoomer imaging. To reduce bias, all images (slices from control and centrifuged mice) were performed randomly in one batch. To localize antisense oligonucleotides in the brain and liver tissues, some images were acquired with SP5 confocal microscope. In this case, fluorescent molecules were excited with the 488 nm line of Argon laser and all acquisition parameters were kept constant. 

### 2.7. Fluorescence Analyses

Several types of fluorescence analyses were double-blindly performed on Nanozoomer images. To evaluate the intensity level of fluorescence, the ndpi files generated by Nanozoomer were converted into tiff images with NDPI software (version 2). The tiff files were opened with Fiji software 2.9.0, and the intensity levels were measured in regions of interest (ROI defined as red circle of 960 µm^2^ in Figures and placed on the hippocampus (HPC), dorsal thalamic nuclei (THAL) and the retrosplenial and ectorhinal cortices (DCx and LCx, respectively) on both hemispheres of the three slices. No filter settings were applied to the images and we checked that the images did not have any saturated dots. The mean of fluorescence was calculated for each mouse and reported in the statistical analysis. A similar analysis was performed in three liver slices. Five ROI were randomly placed on each slice. Moreover, the image analysis of fluorescent spots was performed with QuPath directly on the ndpi files. The software is able to identify and localize fluorescent spots. We empirically determined parameters to segregate fluorescent spots in brain slices from 5 mice (control and centrifuged mice) and we applied these parameters to the project containing the entire sample. The parameters were: pixel size 0.5 µm, background radius 30 µm, median filter radius 0, sigma 1, minimum area 5 µm^2^, maximal area 1000 µm^2^ and threshold 7. The collected data were attributed to experimental groups (2 g vs. 1 g) and compared statistically. The analyses, reported, were performed on the ROI anatomically defined as HPC (hippocampus), THAL (grouping all medio-dorsal and lateral thalamic nuclei), DCx (containing retrosplenial cortices), SoCx (containing somatosensorial cortices) and PirCx (containing piriform cortices). A similar analysis was performed on the left lobe liver slices.

### 2.8. Gene Expression by RT-qPCR

For this experiment, 16 mice were used (8 were exposed to 2 g and 8 to 1 g, as described before). They were anesthetized with isoflurane 5% and decapitated, and the brains were directly frozen and stored at −80 °C. Hippocampus were dissected on ice and placed in 2 mL tubes containing 500 µL of Tri-reagent (MRCgene) and 10 ceramic beads (diameter 1.5 mm). Samples were mashed in a Beadbug6 shaker (Benchmark, 3 cycles, level of speed 4350 and 60 s time). RNA was isolated, following the instruction of the protocol elaborated by MRCgene. The concentration of RNA was measured with Nanodrop (Thermoscientific, Waltham, MA, USA) and adjusted close to 100 ng/µL. The cDNA was produced with the RT-i-script gDNA clear cDNA synthesis kit (Bio-Rad’s reference 1725035), using 100 ng of RNA and following the protocol from the supplier. The qPCR was performed using the endothelial cell contacts by junction M96 (predesigned for use with SYBR green; Bio-Rad’s plate reference 10029202) and the Sso-advanced universal SYBR green PCR kit (Bio-Rad’s reference 1725275). The qPCR was performed with CFX96 thermocycler (Bio-Rad). Samples were allocated randomly in plates, and some of them were tested twice to verify the quality of the experiment. The validation of *Hprt* and *Gapdh* as reference genes was evaluated with CFX Maestro software (Bio-Rad). The analysis of gene expression was performed on *Actb, Actg1, Actn1, Actn2, Actn4, Cdh2, Cdh5, Cldn1, Cldn3, Cldn5, Ctnna1, Ctnnb1, Ctnnd1, Dsp, F11r, Gja1, Gja4, Gja5, Jam2, Jup, Ocln, Tjp1, Tjp2* and *Vim*. The threshold of the regulation by hypergravity on gene expression was chosen at 1.5. To discuss the RT-qPCR results, we checked the brain localization, cell types expressing genes and function of proteins encoded by these genes in endothelial cells using databases: https://www.proteinatlas.org; http://mousebrain.org; http://betsholtzlab.org and https://www.informatics.jax.org (accessed on 26 January 2023).

### 2.9. Statistical Analysis

The data were statistically compared using paired t-tests, non-parametric Mann–Whitney test, or one- and two-way ANOVA with post hoc comparisons when applicable. The normogravity (1 g) is the control condition. The software used was GraphPad Prism V9, which calculated the *p* value as the probability of observing two identical conditions. If *p* < 0.05, the two compared conditions were considered statistically different.

## 3. Results

### 3.1. Effects of Centrifugation on Mice 

The body weight gain, expressed as the difference in weight in a 24 h period (Figure 1), is the difference in body weight measured before and after exposure to centrifugation (2× *g*) or control conditions (1 g). As expected, the exposure to centrifugation induced a decrease in body weight (Figure 1A, *p* < 0.0001). 

More precisely, the decrease in body weight was similar in mice injected with saline solution (Sham) and solution containing fluorescent antisense oligonucleotide directed against angiopoietin-2 (AS) (two-way ANOVA coupled with Sidak post hoc test, interaction *p* = 0.0009; 1 g vs. 2 g: *p* < 0.0001; sham vs. AS *p* = 0.589; Figure 1B). The decrease in body weight gain due to centrifugation was similarly observed in mice injected with dextrans (D40, D70 and D150, one-way ANOVA coupled with Sidak post hoc test, 1 g vs. 2 g: *p* < 0.0001; *p* > 0.05 for comparison of 1 g groups as well as for 2 g groups, Figure 1C). The effects observed in mice injected with AS or dextrans (D40, D70 and D150) were similar (statistical analysis shown in Figure 1C). In conclusion, the injection of fluorescent tracers did not influence the effect of centrifugation on body weight gain. To explain the decrease in body weight, we also measured the food and water consumption. As shown in Figure 1D and 1E, the comparison of food and water consumption, respectively, during the day before the centrifugation with the consumption during the 24 h of centrifugation exposure showed that both food and water consumption specifically decreased in the group exposed to the centrifugation (two-way ANOVA, time x gravity *p* = 0.0001 for both parameters).

The stress was evaluated by the concentration of corticosterone in plasma. The comparison between 1 g and 2 g conditions, including all the samples, did not reveal a variation in corticosterone concentration (Figure 2A, Mann–Whitney test, *p* = 0.255). We also separately analyzed the corticosterone concentration in each experimental group. In mice injected with saline solution (Sham), AS, D40, D70 and D150, the centrifugation had no significant effect on the corticosterone concentration (Figure 2B, one-way ANOVA, *p* = 0.278). In conclusion, centrifugation at 2× *g* did not modify the plasma concentration of corticosterone in mice injected with fluorescent tracers.

### 3.2. Effects of Centrifugation on Extravasation of Fluorescent Dextrans in Brain

The extravasation of dextrans through the BBB were firstly evaluated by the analysis of fluorescence intensities of several brain areas. To minimize local variations, we performed all analyses on slices containing similar anatomical landmarks. The regions of interest were distributed in different cerebral areas (red ROI in thalamus, hippocampus and dorsal and lateral cortices, Figure 3). 

The centrifugation was not able to modify the fluorescent levels in THAL (Figure 3A, Mann–Whitney tests comparing 1 g vs. 2 g conditions for D40, *p* = 0.93; for D70 *p* = 0.29 and for D150 *p* = 0.069), HPC (Figure 3B, Mann–Whitney tests comparing 1 g vs. 2 g conditions for D40 *p* = 0.53; for D70 *p* = 0.76 and for D150 *p* = 0.089) and LCx (Figure 3C, Mann–Whitney tests comparing 1 g vs. 2 g conditions for D40 *p* = 0.50; for D70 *p* = 0.08 and for D150 *p* = 0.16). In DCx, the hypergravity can increase the level of fluorescence only in D70 (Mann–Whitney tests comparing 1 g vs. 2 g conditions for D40 *p* = 0.051; for D70 *p* = 0.040 and for D150 *p* = 0.050; Figure 3D). The differences in D70 fluorescence across brain sections are illustrated in Figure 3E. More marked fluorescence diffusion is observed in the DCx of 2 g-exposed mice. In conclusion, these sets of data analysis suggested that centrifugation significantly increased the presence of D70 in DCx.

### 3.3. Effects of Centrifugation on Extravasation of Fluorescent AS in Liver

We tested the ability of hypergravity to promote the passage of a molecule that can be captured by liver parenchyma cells. To test this hypothesis, we injected mice with fluorescent antisense oligonucleotides and compared the 1 g and 2 g conditions. The same quantification methods used for dextrans were applied on images obtained with Nanozoomer (Figure 4A). 

A significant increase in fluorescence in liver parenchyma was revealed in mice exposed to hypergravity (Figure 4B, Mann–Whitney tests, 1 g vs. 2 g *p* = 0.0291). Moreover, the number of areas containing fluorescence evaluated with QuPath was higher in 2 g in comparison with 1 g (Figure 4C, Mann–Whitney tests, 1 g vs. 2 g *p* < 0.0001). With confocal microscopy, the presence of AS was qualitatively revealed as spots of fluorescence close to vessel walls in the liver parenchyma. Taken together, these results strongly suggest that hypergravity increased the AS extravasation in the liver parenchyma.

### 3.4. Effects of Centrifugation on Extravasation of Fluorescent AS in Brain

The qualitative analysis of images obtained with Nanozoomer and confocal SP5 showed fluorescent spots in the brain parenchyma only in slices from 2 g-exposed mice (Figure 5A and Figure 6A). The confocal images also revealed that these fluorescent spots were more localized in the brain parenchyma close to the vessel walls (Figure 5A, right panel). 

The quantitative analyses of images from Nanozoomer showed an increase in fluorescence level in HPC and DCx due to hypergravity exposure (Mann–Whitney tests, 1 g vs. 2 g in THAL *p* = 0.369, in HPC *p* = 0.033, in DCx *p* = 0.016 and in LCx *p* = 0.265; Figure 5B).

The analysis with QuPath software was used to segregate fluorescent areas from the background in several brain regions (Figure 6A) using the same filtering parameters in both 1 g and 2 g conditions. The analyses confirmed that the exposure to hypergravity increased the number of fluorescent spots in HPC and DCx, but not in THAL (Mann–Whitney tests, 1 g vs. 2 g in THAL *p* = 0.0536, in HPC *p* = 0.0003 and in DCx *p* < 0.0001; Figure 6B). Moreover, it also revealed an increase in the number of fluorescent spots in SoCx and PirCx (Mann–Whitney tests, 1 g vs. 2 g *p* <0.0001 and *p* = 0.0024, respectively; Figure 6B). 

In conclusion, our data suggest that hypergravity induced a BBB leakage able to increase the presence of AS in brain parenchyma. 

### 3.5. Effects of Centrifugation on Expression of Genes Involved in Endothelial Cells Interactions 

Using *Hprt* and *Gapdh* as reference genes, the RT-qPCR analysis of the expression of genes involved in the regulation of endothelial cells interactions revealed that *Gja4*, *Ctnnd1* and *Actn1* were upregulated. *Cdh2* was downregulated more than 2-fold and *Ocln*, *Actn2*, *Jup*, *Actn4*, *Tjp2* and *Gja1* were downregulated between 1.5- and 2-fold (Figure 7). The expressions of *Actb, Actg1, Cdh5, Cldn1, Cldn5, Ctnna1, Ctnnd1, Dsp, F11r, Gja5, Jam2, Tjp1* and *Vim* were considered not altered (less than 1.5-fold modification), and *Cldn3* appeared not expressed. The names, functions and cell types expressing these genes are summarized in Table 1 and Appendix A.

## 4. Discussion

In the present study, our results suggest that hypergravity induces an increase in BBB permeability, allowing the passage of antisense oligonucleotides as well as dextran from blood to brain parenchyma. Moreover, the RT-qPCR experiments suggested an alteration in the expression of genes involved in endothelial cell junctions. 

In a ground model of hindlimb unloaded animals without [56] or in combination with radiation [57], as well as during spaceflight [58], the BBB was altered, suggesting that vestibular regulations were involved. As reviewed recently, the increase in gravity by centrifugation modifies vestibular function and induces motion sickness [59]. Our experiments confirm the decrease in body weight generated by centrifugation [18,60]. It is linked with the decrease in food intake [61], and probably linked to vestibular impairments [18,62].

Hypergravity exposure at 2 g increases the corticosterone concentration when it is measured during the first hour of exposure [63]. The increase in the hypergravity level can transiently increase the plasma corticosterone level [64]. Nevertheless, as our data showed, after 24 h of weak exposure at 2 g, the corticosterone levels were not altered in the first hour following the stop of the centrifuge [65]. The stress induced by the centrifugation is controversial and probably depends on the design of the centrifuge and experimental procedure with animals [18,27,63]. Moreover, our data showed a large spread of individual values of corticosterone concentration, confirming other studies [18,23,65]. 

In motion sickness, the relationship between brain and intestinal functions were known and clearly demonstrated, including microgravity and hypergravity models [66,67,68]. The most probable link is hypophagia. In mice and rats, the decrease in food intake was observed at the beginning of the 2 g exposure (first two days) and depended on the vestibular organ [18,69]. 

The hypophagia could have several causes, including: 1. modifications in microbiota [70] that can decrease the gastric acid synthesis [71], 2. metabolism dysregulation, such as decreases in leptin and insulin plasma concentrations [60] and 3. modifications in the expression of the starvation-induced genes [72]. Moreover, the serotonin pathways are involved in this phenomenon [69,73]. In conclusion, our results also confirm that the hypophagia induced a decrease in body weight. This is more related to the hypergravity and not related to an increase in corticosterone levels [30,60,62,65]. 

Fluorescent polysaccharides such as dextrans are safe at low concentrations, available in sizes from 3 to 2000 kDa. They can be used to study BBB permeability [74,75,76] and to determine the size of a BBB leak [77,78,79]. After 24 h exposure to 2 g hypergravity, our results demonstrated that 70 kDa dextran can be exported in cortex parenchyma, but not 40 or 150 kDa dextran. The lower molecular weight dextrans, the faster they are excreted. In fact, in less than one hour, dextrans between 30 and 40 kDa were excreted in urine, whereas the 62 kDa dextran was always present in the blood circulation and not highly present in urine [80]. This suggests that after 24 h, the 40 kDa dextran would be excreted. Thereby, the BBB leak required more than one hour of hypergravity exposure, confirming our previous data showing that short exposure (1–9 min at 5 g) was not efficient in destabilizing the BBB [65]. Because we cannot exclude an alteration in urine excretion of dextran in the hypergravity context, our data should be completed by the evaluation of the kinetics of dextran excretion in centrifuged mice.

Because of the molecule shape, 150 kDa dextran was unable to flow from the circulation to the tissues in physiological conditions [80]. Our results showed that the BBB leak is not sufficient for 150 kDa dextran extravasation, suggesting that this leak was not comparable to the BBB disruption induced by stroke or acute hypertension [81]. In our previous study [65], the extravasation of IgG (around 150 kDa) was measured, suggesting that the nature of the molecule is also a crucial parameter. Moreover, our data showing the extravasation of antisense oligonucleotide in the cortex and hippocampus confirm that the BBB properties depend on the brain areas and the chemical nature of the markers [82,83,84]. In conclusion, our results showed an increase in the transfer of fluorescent molecules from blood to tissues, suggesting a global modification in effluxes due to hypergravity.

To assess the alterations in the BBB in centrifuged mice, we focused the molecular investigation on gene expression using a set of primers targeting consensual genes involved in BBB efficacy. As reviewed recently [85,86], all of the proteins encoded by the genes studied here are involved in the scaffoldings required to maintain endothelial cell interactions to create the BBB, as well as in the initiation of angiogenesis and/or vascular repair. The database queries concerning the expression level in non-neuronal cells of the brain indicated that the proteins encoded by studied genes are also expressed in endothelial cells, but not exclusively (Appendix A). As expected, the modifications in gene expression are related to the durations of centrifugation and the levels of hypergravity, as suggested by the comparison between this current study and the RNAseq performed previously on the same device and the same mouse strains [29]. Moreover, the regulation of gene expression is not comparable to acute and chronic stress (Appendix A). Globally, the observed modifications could be interpreted as a specific dysregulation of gene expression that can alter the turnover and replacement of proteins involved in BBB efficacy as observed in BBB disruption models such as stroke, middle cerebral artery occlusion or hypoxia.

## 5. Conclusions

This work suggests that the modification in gravity, which is accompanied by a modification in the vestibular functions, leads to an alteration in the BBB via a modification in the expression of genes which code the proteins in the junctions between the endothelial cells. As now studied, an alteration in the BBB, and not its destruction, allows the passage of molecules defined by their sizes and their chemical natures. Our work insists on this point; an alteration in the BBB is characterized according to the means of study, i.e., markers and measurement methods. This can be considered in two antagonistic ways, either as a minimally invasive physical means of crossing the BBB by molecules of therapeutic interest or, on the contrary, as something deleterious that can be found in the pathology of alterations in vestibular functions during spaceflight. The most important limit of this study is that the RT-qPCR was performed on RNA extracted from whole brain, and the query of hipposeq.janelia.org indicated that we cannot exclude the alteration in molecular scaffolding of synapses also implicating these genes. Finally, our study can be considered an extension of studies relating to the effectiveness of molecules to modulate the passage across the BBB. In a hypergravity context, but also in other models of alteration in vestibular functions, the transduction pathways involved in alterations in the BBB should also be investigated. For example, the angiopoietin-2 pathway is crucial for endothelial cell disassembly [87], and GPCR internalization in endothelial cells [88] should be considered in the context of centrifugation. The last topic that we can investigate is the effects of gravity modulation on angiogenesis, which is required to renovate the endothelium and form new brain capillaries. In fact, experiments on cultured endothelial cells have suggested that hypergravity reduces their capacity to form tubes and alters their responses to angiogenic factors [48,49,50,51]. In centrifugation as well as during parabolic flights, the in vivo responses to angiogenic factors have not yet been investigated. Moreover, it has been shown that during the takeoff and landing of a space module (BION-M 1), hypergravity induces cardiovascular changes [89]. More experiments should be conducted to precise how these cardiovascular changes can modify the structure of the BBB and neurovascular unit functions. To restore physiological functions after spaceflight or bed-rest in humans, a daily sequence of short exposure to centrifugation close to 2× *g* has been proposed. It is crucial to verify if this protocol has any effect on the BBB. Recently, biomarkers of BBB alteration have been listed [90], and their investigation in the spaceflight context should be performed. Finally, centrifugation can be considered to potentiate vectorization and should be used to investigate cell functions with antisense oligonucleotides in pathophysiological contexts.

## Figures and Tables

**Figure 1 cells-12-00734-f001:**
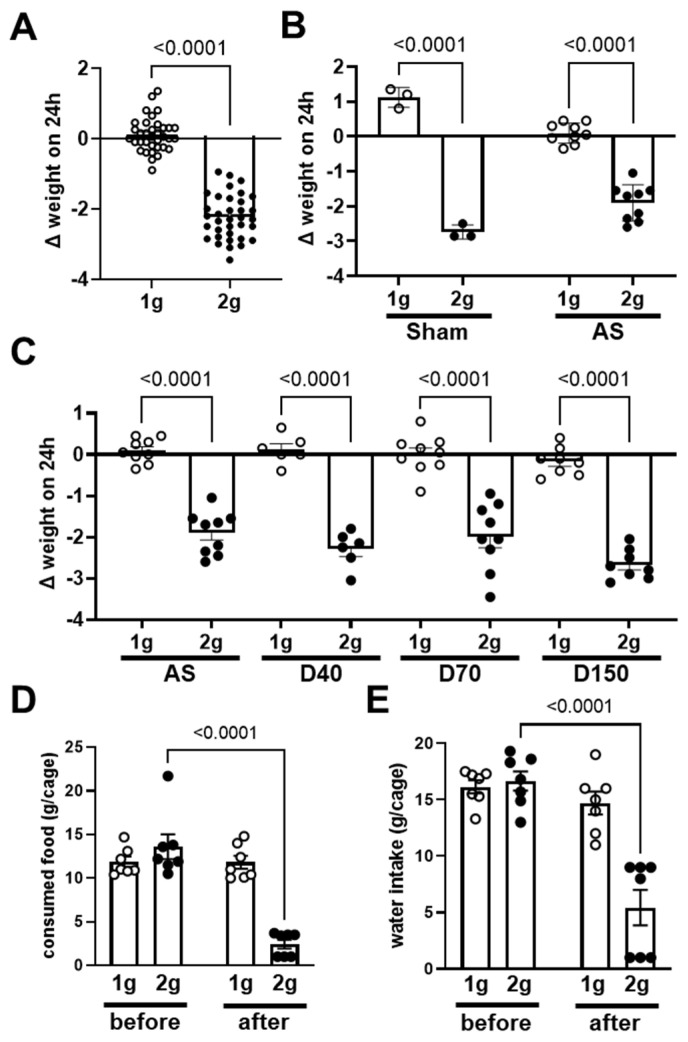
Variation in body weight and food and water consumption in normogravity and hypergravity. (**A**) Variation in body weight (Δ weight on 24 h expressed in grams) in 24 h in all mice exposed to both normogravity (1 g, circle) and hypergravity (2 g, point) conditions (n = 35 per condition). (**B**) Variation in body weight of mice injected with saline solution (Sham, n = 3 per condition, circle) and antisense oligonucleotide (AS, n = 9 per condition, point). (**C**) Variation in body weight of mice injected with antisense oligonucleotide (AS, n = 9 per condition), dextrans D40 (n = 6 per condition), D70 (n = 9 per condition) and D150 (n = 8 per condition). *p* values < 0.05 are precise. (**D**) Food and (**E**) water consumption during the 24 h; before the centrifugation (before) and after the period of centrifugation (after) in both 2× *g* and in 1 g groups. The consumption of food and water are expressed in g per cage, each cage containing three mice (n = 7 cages per group).

**Figure 2 cells-12-00734-f002:**
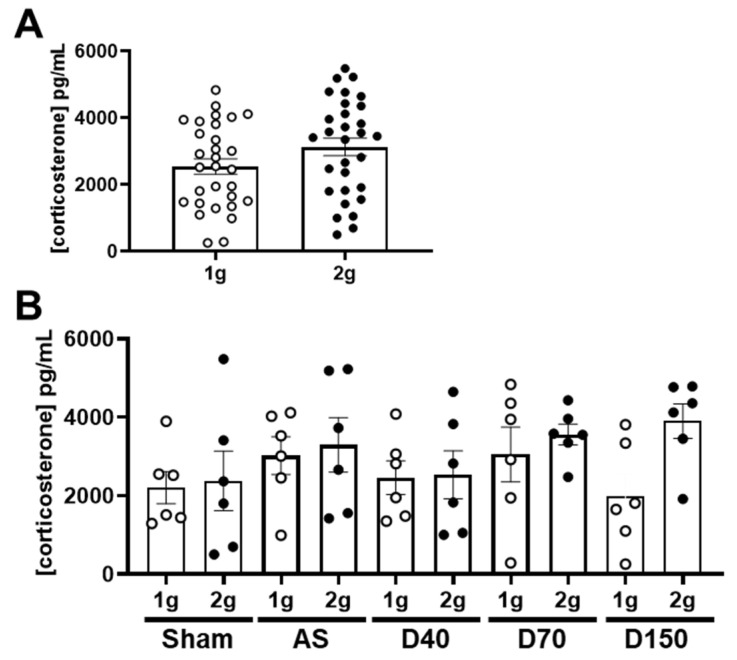
Effects of centrifugation on corticosterone concentration. (**A**) Corticosterone concentration (in pg/mL) in both normogravity (1 g, circle) and hypergravity (2 g, point) conditions (n = 30 per conditions). (**B**) Corticosterone concentration (in pg/mL) in both normogravity (1 g) and hypergravity (2 g) conditions in Sham-, AS-, D40-, D70- and D150-injected mice (n = 6 per conditions).

**Figure 3 cells-12-00734-f003:**
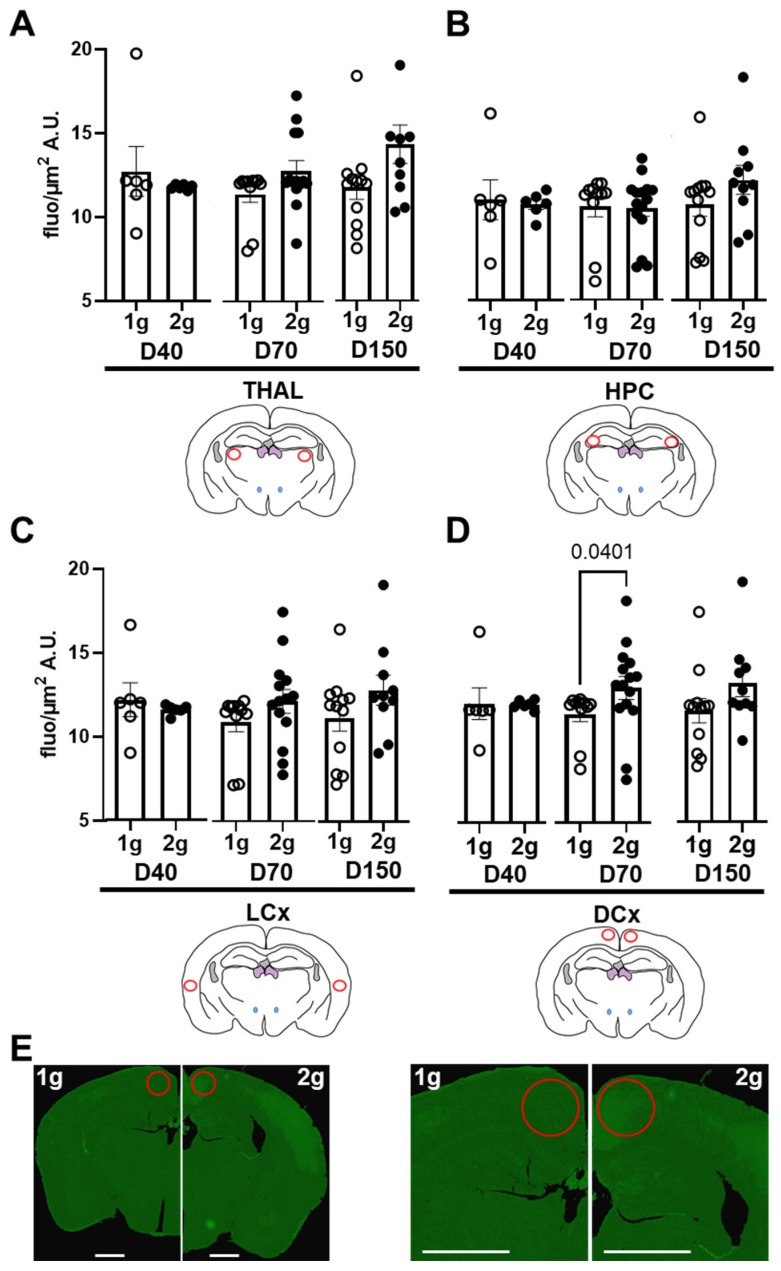
Effects of centrifugation on extravasation of fluorescent dextrans in brain. Measures were performed on Nanozoomer images from (**A**) thalamus (THAL), (**B**) hippocampus (HPC), (**C**) dorsal (DCx) and (**D**) lateral (LCx) cortices. Fluorescence levels were used as an index of extravasation. They were expressed as grey level (arbitrary unit, A.U.) per surface (µm^2^) of the D40 (n = 11), D70 (n = 25) and D150 (n = 22), measured in both normogravity (1 g, circle) and hypergravity (2 g, point) conditions. Red circles on schema of the brain slices represented the ROI used for the analyses, and the p values < 0.05 are indicated. (**E**) Typical images obtained with Nanozoomer of brain slice from D70-injected mice exposed to normogravity (1 g, left) and hypergravity (2 g, right). The right panel is a magnification of the left one. Red circles indicate the ROI used for fluorescence measurements. Scale bar represents 1 mm.

**Figure 4 cells-12-00734-f004:**
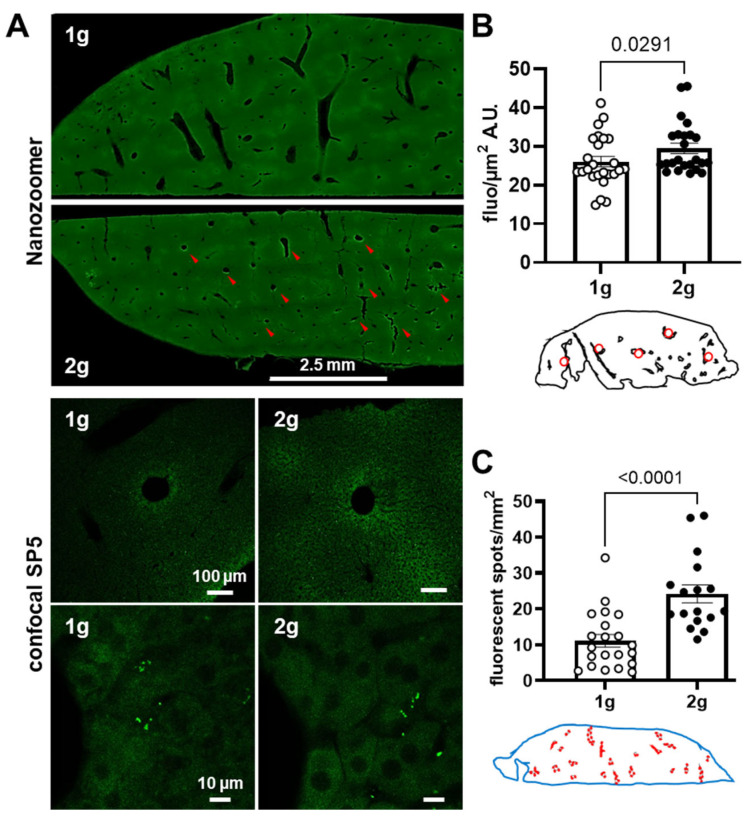
Effects of centrifugation on extravasation of fluorescent AS in liver. (**A**) Typical images of liver slices from mice exposed to normogravity (1 g) and hypergravity (2 g) obtained with Nanozoomer (upper panel) and confocal SP5 (lower panels, with two different magnifications). Red arrows indicate AS fluorescent spots. (**B**) The fluorescent level, expressed as grey level (A.U.) per surface (µm^2^) of the AS in liver (n = 24), was used as an index of extravasation. Red circles on schema of the liver slices represented the ROI used for the analysis. (**C**) Density of fluorescent spots (number of spots per mm^2^) used as an index of AS inclusion into the liver tissue. The analysis was performed in the whole slice (blue shape) and red circles on the schema indicated the localization of spots. The 1 g and 2 g conditions are associated with white and black points, respectively, and the *p* values < 0.05 are indicated.

**Figure 5 cells-12-00734-f005:**
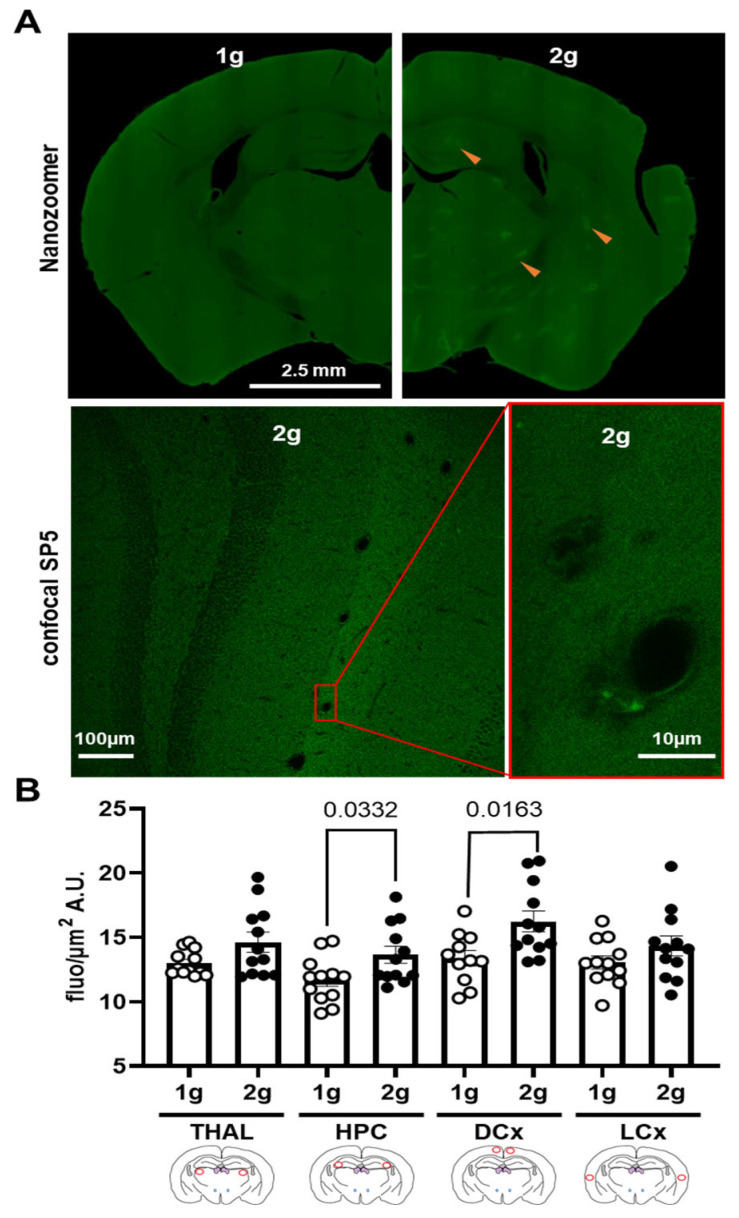
Effects of centrifugation on extravasation of fluorescent AS in brain. (**A**) Typical images of brain slices from mice exposed to normogravity (1 g, upper panel, left) and hypergravity (2 g, upper panel, right) obtained with Nanozoomer. Red arrow indicates fluorescent spots. A focus on the hippocampus area of the brain slice from mice exposed to hypergravity (2 g, lower panel) is illustrated by confocal SP5 images obtained at two different magnifications (right panel was an enlargement of the image on the left; corresponding to the red rectangle). (**B**) Fluorescence level expressed in grey level (A.U.) per surface (µm^2^) of the AS (1 g, n = 12 vs. 2 g, n = 12) in thalamus (THAL), hippocampus (HPC), dorsal (DCx) and lateral (LCx) cortices used as an index of extravasation. Red circles on schema of the brain slices represent the ROI used for the analyses; the 1 g and 2 g conditions are associated with white circle and black points, respectively, and the *p* values < 0.05 are indicated.

**Figure 6 cells-12-00734-f006:**
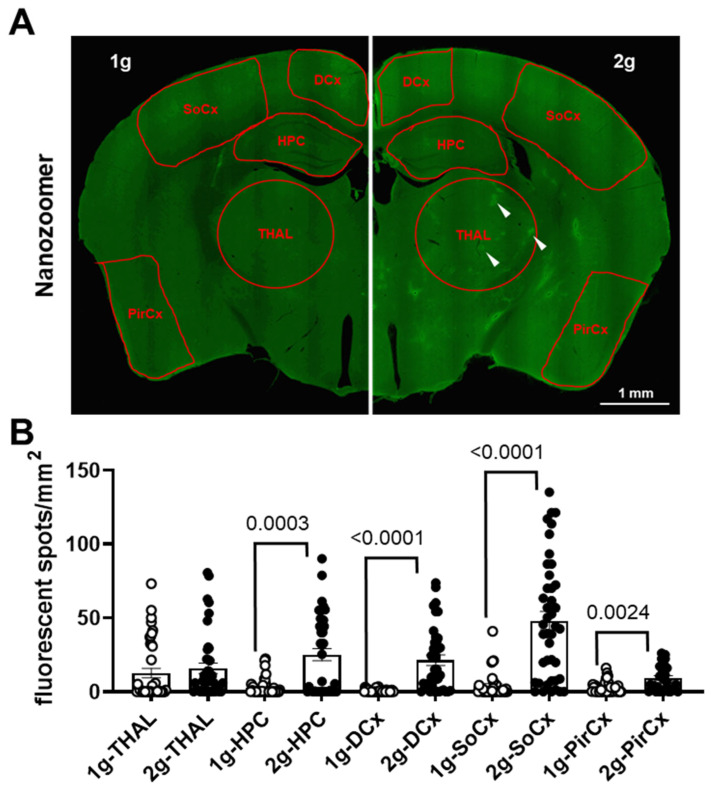
Density of fluorescent spots in brain. (**A**) The brain regions used for the analysis were delimited by red shapes on Nanozoomer image (1 g on the left and 2 g on the right). These red ROI were placed on the thalamus (THAL), hippocampus (HPC), dorsal cortex (DCx), somatosensorial cortex (SoCx) and piriform cortex (PirCx). The white arrows indicate some spots with high levels of fluorescence. (**B**) Density of fluorescent spots (number of spots per mm^2^) was used as an index of AS diffusion in the brain parenchyma. The 1 g and 2 g conditions are associated with white circles and black points, respectively, and the *p* values < 0.05 are indicated.

**Figure 7 cells-12-00734-f007:**
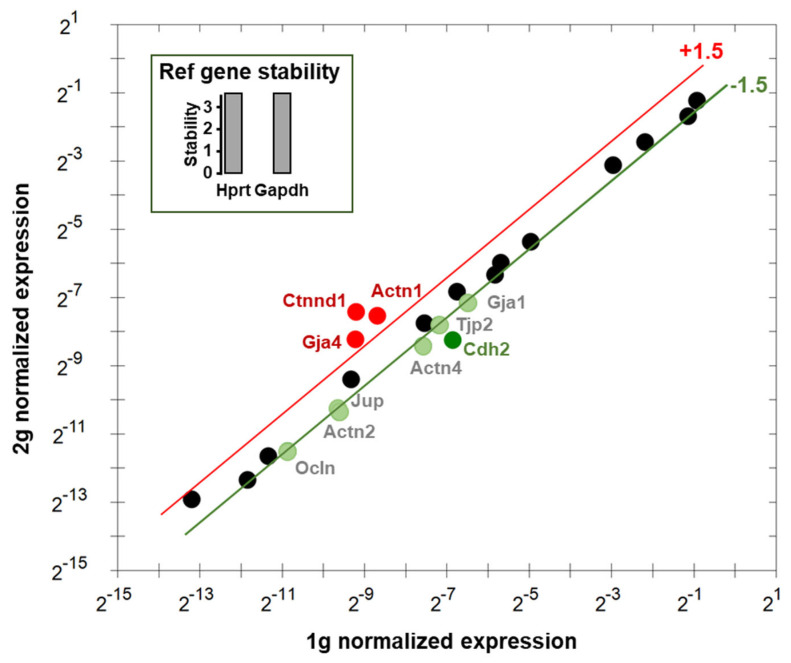
Relative expression of genes involved in endothelial cell contacts. The gene expressions in 1 g and 2 g were compared using *Hprt* and *Gapdh* as reference genes (in inset). The upregulated and downregulated genes are indicated in red and green, respectively. Genes indicated in black are considered not affected.

**Table 1 cells-12-00734-t001:** Relative expression and functions of genes modulated by 2 g.

Gene	Name	Relative exp. ^#^	Function of the Genes
*Actn1*	Actinin α1	2.24	- Adherens-type junctions, binding actin to the membrane.
*Actn2 **	Actinin α2	−1.67	- Adherens-type junctions, binding actin to the membrane.
*Actn4*	Actinin α4	−1.78	Probably involved in vesicular trafficking.
*Cdh2*	Cadherin 2	−2.61	- Ca^2+^-dependent cell adhesion molecule; development of the nervous system; may regulate dendritic spine density.
*Ctnnd1*	Catenin δ1	3.46	- Key regulator of cell–cell adhesion; regulates gene transcription and activity of Rho family GTPases and downstream cytoskeletal dynamics.
*Gja1*	Gap junction protein α1	−1.58	- Gap junctions, intercellular channels for the diffusion of low molecular weight materials; major protein of gap junctions in the heart with crucial role in the synchronized contraction of the heart.
*Gja4*	Gap junction protein α4	2.00	- Gap junctions, intercellular channels for the diffusion of low molecular weight materials, element of the connexons.
*Jup*	Junction plakoglobulin	−1.52	- Major cytoplasmic protein integrated in both desmosomes and intermediate junctions; forms complexes with cadherins and desmosomal cadherins.
*Ocln*	Occludin	−1.54	- Membrane protein required for formation and (cytokine-induced) regulation of the tight junction paracellular permeability barrier; induces adhesion when expressed in cells lacking tight junctions.
*Tjp2*	Tight junction protein 2	−1.52	- ZO-2 (zonula occluden) a component of the tight junction barrier in epithelial and endothelial cells.

* Very low expression in non-neuronal cell types (database: MGI.org and genecards.org). ^#^ Relative exp., expressing relative expression as 2 g-fold change against 1 g.

## Data Availability

Data are available upon request from the corresponding authors.

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
