# Peer review of "Hypergravity Increases Blood–Brain Barrier Permeability to Fluorescent Dextran and Antisense Oligonucleotide in Mice"

_cells, 2023, doi:10.3390/cells12050734_

Round 1

Reviewer 1 Report

The purpose of the present study was to examine the effect of hypergravity on the blood-brain barrier and the regulation of expression of genes involved in junctions between endothelial cells using centrifugation. This study is interesting, and I believe that the experiment was performed carefully the findings of the present study provide some important information limited in this research area. However, the motivation of this study is unclear. My understanding is that centrifugation is useful as a countermeasure against microgravity-induced motion sickness (vestibular dysfunction etc.) for astronauts but we should know the effect of hypergravity on the human body and physiological function (BBB etc.). If so, the authors should provide the necessity for the limited knowledge for using centrifugation as a countermeasure for long-term microgravity. I understand that important to know the effect of hypergravity as a change in gravitational condition on the human body but at least, the authors should provide a clear rationale and research impact of the present study. It is not clear. For example, it is unclear why the stimulation of hypergravity is 2 g for 24 h, I think (astronauts use centrifugation for 24 h continuously? If so, please explain why).  It is possible that this stress rather than gravity change may affect physiological factors. If this study is for confirming the validity of the countermeasure for astronauts, is 24 h too long? Also, for astronauts, the use of centrifugation may be under microgravity. Anyway, the authors should explain the study implications more clearly for readers although I believe that the data is important in this research area.

Author Response

We thank the both reviewers for their comments that help us to increase the scope of our MS.

Reviewer 1, your comment told us that we needed to rethink and rephrase the introduction to be more incisive about the importance of our work in the field of space motion sickness induced by gravity changes. We have therefore modified the introduction in this direction, hoping to have made the text clearer (see modifications in blue, lines 2-6). For that, we add some references on the crucial contribution of space biology in this field, and the use of centrifugation to reproduce the space motion sickness observed in the first days of the spaceflights. We suppress some sentences about other biological effects of centrifugation because they cause confusion in our purpose. Finally, we add sentence in conclusion about the requirement for other studies using short and daily centrifugation exposures to reproduce protocols suggested and tested as countermeasures after spaceflight or bed-rest (see modifications in blue, lines 461-463).

We would like also indicate that hypergravity studies could be distinguished in two groups: 1/ studies reproducing the severe acceleration measured in the war planes for example (close to 9g during few seconds or minutes) and 2/ studies using hypergravity to evaluate it as countermeasure to reverse biological effects due to spaceflight (2g-3g during 10-20 min/day during weeks in human and continuously during days or weeks in animal models). In the first kind of studies, the high level of hypergravity is always associated to a drastic increase of stress markers whereas in the second case the levels of corticosterone in animal models or cortisol in human were similar to those measured in control conditions. This point is clearly not the goal of our purpose but it is important to notice that the level of gravity is a fundamental parameter, continuous exposures during days to 4g and more are deleterious for life of animal models.

Reviewer 2 Report

I read the manuscript with interest. The topic and the experimental design are interesting and important. However, a careful revision is necessary before publication.

Major points

What is the novelty of the findings of this article? In the discussion part several statements and observation are mentioned, but it is not clear whether these are the authors' new results or previously published data by other research groups. The discussion section should be reorganized or rephrased.

The cerebral extravasation of 70kDa dextran is described, but according to the authors the 40 kDa dextran was excreted in the urine. This seems to be a speculation, no experimental data behind it. It would be logic, if 40 kDa dextran also becomes extravasated, but if it is never present in plasma, then it was not a good choice as a tracer.

Why was the retro-orbital route of administration selected for the tracers? How was it performed exactly?

The body weight and food and water consumption were reduced after the 24h centrifugation at 2g. However, the effect of this stimulation on the gastrointestinal tract was not analysed. Please add some sentences and refrences about it.

Alauzet C, Cunat L, Wack M, Lozniewski A, Busby H, Agrinier N, Cailliez-Grimal C, Frippiat JP.Hypergravity disrupts murine intestinal microbiota.

Sci Rep. 2019 Jun 28;9(1):9410. doi: 10.1038/s41598-019-45153-8. Yoon G, Kim HS.Gastric acid response to acute exposure to hypergravity.Oncotarget. 2017 Jan 3;8(1):64-69. doi: 10.18632/oncotarget.13969. 

Minor points

line 44 nutrients

line 104 cryopreserved at room temperature??

line 137 to quantitate

Beside Fig 7, a table containing the tested genes should be added with the relative expression compared to the 1g control group and with the description of the functions of the genes.

At the Funding section the grant numbers are missing.

Please add this review paper about BBB opening:

L.Bors and F. Erdő, Overcoming the Blood–Brain Barrier. Challenges and Tricks for CNS Drug Delivery,Sci. Pharm. 2019, 87(1), 6; https://doi.org/10.3390/scipharm87010006

Author Response

Please find our answers to your question in the joint file

Round 2

Reviewer 1 Report

Thank you for your revision. I believe that this manuscript was improved.

No further comments.

Reviewer 2 Report

I accept the manuscript for publication in the current form.